# Current and Future Perspective of Devices and Diagnostics for Opioid and OIRD

**DOI:** 10.3390/biomedicines10040743

**Published:** 2022-03-22

**Authors:** Naveen K. Singh, Gurpreet K. Sidhu, Kuldeep Gupta

**Affiliations:** 1Department of Biomedical Engineering, Pennsylvania State University, State College, PA 16803, USA; 2Department of Obstetrics, Gynecology, and Reproductive Sciences, School of Medicine, University of California San Diego, La Jolla, CA 92093, USA; gksidhu@health.ucsd.edu; 3Russell H. Morgan Department of Radiology and Radiological Science, Johns Hopkins University, Baltimore, MD 21287, USA

**Keywords:** opioids, OIRD, diagnostics, sensor, healthcare

## Abstract

OIRD (opioid-induced respiratory depression) remains a significant public health concern due to clinically indicated and illicit opioid use. Respiratory depression is the sine qua non of opioid toxicity, and early detection is critical for reversal using pharmacologic and non-pharmacologic interventions. In addition to respiratory monitoring devices such as pulse oximetry, capnography, and contactless monitoring systems, novel implantable sensors and detection systems such as optical detection and electrochemical detection techniques are being developed to identify the presence of opioids both in vivo and within the environment. These new technologies will not only monitor for signs and symptoms of OIRD but also serve as a mechanism to alert and assist first responders and lay rescuers. The current opioid epidemic brings to the forefront the need for additional accessible means of detection and diagnosis. Rigorous evaluation of safety, efficacy, and acceptability will be necessary for both new and established technologies to have an impact on morbidity and mortality associated with opioid toxicity. Here, we summarized existing and advanced technologies for opioid detection and OIRD management with a focus on recent advancements in wearable and implantable opioid detection. We expect that this review will serve as a complete informative reference for the researchers and healthcare professionals working on the subject and allied fields.

## 1. Introduction

The prevalence of opioid-induced respiratory depression (OIRD) remains a public health concern due to both clinically indicated and illicit opioid use. The number of drug overdose deaths almost tripled between 1999 and 2015, primarily due to opioid overdose. The death toll from opioid overdoses in the United States is approximately 115 per day [1]. Compounds that bind to opioid receptors are referred to as opioids. Typically, the term opioid refers to natural alkaloid derivatives of opium poppies such as morphine and codeine. Moreover, semi-synthetic opioids have also been synthesized from natural opioids, including oxycodone from thebaine and heroin from morphine. Methadone, fentanyl, and propoxyphene are synthetic opioids. While opioids have known risks, they remain one of the most common types of treatment for postoperative pain as well as for certain chronic pain conditions. According to a recent study, hospital-acquired OIRD ranges from 0.1% to 37% after controlled opioid administration, with improper monitoring accounting for almost one-third of cases [2]. The rate of overdose deaths in outpatient settings for patients receiving opioids for non-cancer chronic pain has been observed to range from 0.2–1.8% yearly [3]. Furthermore, those people who use drugs (PWUD) account for a significantly higher proportion of opioid overdose deaths compared to those treated for pain with medically indicated chronic opioids [4]. Opioid toxicity is characterized by respiratory depression and its early identification and diagnosis are crucial for administering effective treatment. Opioids reduce respiratory function by binding opioid receptors in brainstem respiratory centers and other sites along the central and peripheral nervous systems. Under normal circumstances, hypoxia and hypercarbia increase respiratory effort and rate. However, opioids dull both of these physiological responses. As a result, there is a pressing need for low-barrier technologies that can detect opioid excess in vivo, identify its downstream effects (such as OIRD), and enable rapid interventions to prevent harm from opioid toxicity. In the present review, we discuss existing and emerging technologies for detecting opioids and diagnosing opioid-induced respiratory distress (OIRD) in vivo, as shown in Figure 1.

## 2. Mode of Action of Opioids in Pain Suppression and OIRD

Opioids are very effective analgesics and are used for the treatment of severe pain but in the past few decades, there has been an upsurge in the abuse and overdose of opioids. Opioids can be classified into exogenous (morphine, heroin, etc.), administrated from outside, and endogenous (enkephalins), released at a specific site of action by immune cells. Both kinds of opioids recognize the same type of receptors, and it would be rational to observe similar effects on the central nervous system. However exogenous opioids are associated with various side effects [5].

Endogenous opioids constituting peptides or polypeptides enkephalins, endorphins, endomorphins, dynorphins, and nociceptin are components of natural internal pain modulating system. The natural pain modulation is mediated by an interplay between endogenous opioids, opioid receptors, microglia, and neurons. The exogenous opioids accomplish analgesia by activation of cell membrane opioid receptors, leading to interaction with the pain signaling pathway in several ways. Firstly, the activated opioid receptors inhibit the release of bradykinins leading to an anti-inflammatory response. The activated opioid receptors also inhibit the peripheral nociceptors (a sensory receptor that detects painful stimuli) [6]. Secondly, the activated opioid receptors in the dorsal horn of the spinal cord inhibit the voltage-dependent calcium channels leading to blockage in the release of neurotransmitters from afferent dorsal root neurons [7]. Thirdly, the activated opioid receptors in the dorsal horn postsynaptic neurons prevent the propagation of action potential along afferent ascending spinal thalamic tracts by opening potassium channels leading to hyperpolarization of the postsynaptic membrane [8]. These interactions lead to the combinatorial effect of dampening of pain signals through the ascending spinal pathways. The activated opioid receptors inhibit the synapses in the medulla, pons, ventral tegmental area, ventral caudal nucleus of the thalamus, and the cerebral cortex, further reducing pain signals. The inhibitory tone of the brainstem that inhibits descending pathways is also inhibited by opioid receptors, leading to decreased pain signals to brain [9].

The cellular mechanism of opioid-mediated pain modulation involves four major receptors: µ, δ, κ, and the nociceptin opioid receptor (opioid receptor-like receptor) also called MOR, DOR, KOR respectively, as per standard terminology. These four receptors are encoded by a single gene each, but the splice variants from each of the genes lead to different subtypes such as MOR1, MOR2, MOR3, DOR1, DOR2, KOR1, KOR2, NOR1, NOR2, etc. These receptors are G-protein coupled receptors in the cell membrane, present in the central nervous system, peripheral nervous system, immune system, and gastrointestinal system. The MOR1 receptor is most commonly linked to analgesia and dependence. MOR2 receptor is associated with respiratory depression, as well as physical dependence. The binding of the opioid to its receptors induces conformational changes in the latter leading to activation of G coupled protein, which in turn inhibits adenyl cyclase and hence depletion of cAMP. In other words, the binding of opioids induces depletion of cAMP and protein kinases, which in turn causes modulation of gene transcription and increased activity of sodium-potassium ATP pump leading to hyperpolarization of nerve axons as well as synapses and dampening of neuronal signals of pain [10]. The beta (β) and gamma (γ) subunits of G nucleotide-binding protein when released upon opioid binding cause inhibition of voltage-dependent calcium channels on presynaptic neurons and activation of G protein activated inwardly rectifying potassium channels [10]. The inhibition of voltage-dependent calcium channels leads to a decrease in the release of presynaptic neurotransmitters including glutamate, norepinephrine, serotonin, acetylcholine, and substance P [11]. The cumulative effect is hyperpolarization of the postsynaptic membrane and hence dampening of the signal via the spinal cord (Figure 2).

The opioid-induced respiratory depression (OIRD) characterized by a drastic decrease in frequency and regulation of inspiratory rhythm, originates from preBötzinger Complex (preBötC) and is mediated via, (i) μ-opioid receptor (MOR) -dependent mechanism, and (ii) μ-opioid receptor (MOR) -independent mechanism [12]. The binding of opioids with MOR leads to the hyperpolarization of neurons followed by suppression of intracellular cAMP concentration which leads to activation of serotonin receptor (5HT1α) and glycine receptor (GlyR α 3-glycine receptor). These series of cascade mechanisms combined or independently reduce the neuron excitability of respiratory centers (Figure 3) within the pons and medulla, which are accountable for respiration rate via controlling the pneumotaxic, apneustic area and respiratory muscles (inspiration and expiration) that maintain the tidal volume [13]. The hyperpolarization of MOR expressing preBötC neurons leads to a decline in pre-inspiratory spiking as well as suppression of excitatory synaptic transmission, leading to disruption of rhythmogenesis [11].

## 3. Detection of Opioids

The levels of opioids have been studied in most tissues, secretions, and excreta released from the body. Some biological samples are tough to handle due to their inherent complexity and require additional preparation steps before testing. The choice of a biological sample for opioid detection depends on the available infrastructure and resources. Among a wide variety of samples, urine is preferred because it offers an inexpensive and non-invasive approach, and the metabolites tend to concentrate over time in the urine. The collection of saliva (oral fluid) is also noninvasive, but it is more expensive due to the processing steps involved. Opioid concentrations and detection windows in saliva more closely resemble blood/serum profiles than in urine. However, not all kinds of opioids are detectable in saliva. It should be noted though that both saliva and urine samples are susceptible to adulteration or substitution by donors. For the correlation of signs and symptoms with drug concentrations in an acute setting, blood (serum or plasma) is the preferred specimen. The collection of blood is an observed procedure and reduces the odds of specimen substitution or adulteration. In recent years, the remarkable advancement in the field of diagnosis offers the analysis of unconventional drugs from the sweat [14]. Although, there is a long time lag in its emergence in sweat after the dose administration [15]. Hair samples, meconium, and umbilical cord tissues can be used to demonstrate chronic exposure/use [16]. A variety of methods are available for opioid detection, which broadly can be classified into the current clinical methods, conventional methods, and advanced point of care methods. Defining clear boundaries between current clinical and conventional methods is difficult since any of these can be used interchangeably depending on the available resources (sample, operator, instruments, etc.) and the type of required relevant information.

### 3.1. Clinical Testing of Opioids

Clinical testing for opioids plays a vital part in the detection and control of drug abuse. The immunoassay, chromatography, spectrophotometry, or a combination of these techniques are commonly used methods for opioid testing. Among these, immunoassay (or rapid detection kits, RDTs) based on the principle of lateral flow immunochromatographic assay is most frequently used. In this method, a nanoparticle-tagged antibody specifically detects opioids over a chromatographic strip and aggregates over captured antibodies, leading to a colorimetric response. The detection range and limit of detection (LOD) of commonly available RDTs in the market are approximately 0.015–6 and <0.3 µM, respectively in spiked urine and saliva samples [17]. These RDTs are simple, quick, inexpensive, portable, and require a small sample volume (µL), but on the other hand suffer from drawbacks such as being non-quantitative, poor performance in a hot and humid climate, and variability of results [18]. The RDTs have great importance in the clinical setting but confirmatory testing is always recommended to elucidate the results. To confirm positive RDT results, further testing is usually done by gas or liquid chromatography (GC or LC) and mass spectrometry (MS) [19,20]. The MS has outstanding specificity or capability to separate individual components from a mixed sample and provide a distinctive pattern of mass to charge (*m*/*z*). The “DE Tector flex” (Bruker, Billerica, MA, USA) and “RoadRunner” (Bruker) are portable MS devices, and are commonly used by law enforcement agencies to detect opioids and their other derivatives at the nanogram sensitivity level [21]. Recently, a validation study was performed with a portable “Torion T9” GC-MS system (Perkin Elmer, New York, NY, USA), signifying the potential to detect the opioid at very low concentrations in POC (point of care) setting and avoiding the need for off-site confirmation [19]. While GC/LC-MS is currently the gold standard for drug testing, it has several drawbacks, including lengthy analysis times (results may take several days to arrive since they are sent to centralized testing facilities), high costs, and the requirement of skilled operators [22].

### 3.2. Conventional Laboratories Techniques for Opioids Testing

Over the last decade, sophisticated and automated instrument-based methods are developed for the detection of opioids from complex (biological, mixed) samples. These types of methods endowed by smart design and data analysis tools (chemometric) fulfill unmet detection needs [23]. Here are a few techniques commonly used for the detection of opioids.

#### 3.2.1. Raman Spectroscopy

Raman spectroscopy is a broadly used technology for detection applications even though it is suffering from weak plasmonic signal intensities. The signal may be enhanced by introducing an analytic molecule near-source or at the rough noble metal surface. This phenomenon is termed as surface-enhanced Raman spectroscopy (SERS). The sensitivity of SERS based system is very good and has the potential to be used as a point of care method. The detection of opioids using SERS is possible from a wide variety of samples of forensic importance. Alharbi and coworkers have developed a novel approach in which tramadol is adsorbed on the surface of silver hydroxylamine nanoparticles through ionic interactions from the sample, which leads to colloidal aggregation by using sodium chloride at neutral pH [24]. This aggregation results in tramadol concentration-dependent SERS spectral peak area at 993 cm^−1^. The detection range of this method was reported to be 0.05–10, 0.12–8.75 mM in spiked water and artificial human urine sample, respectively, which cover the typical levels present in a drug user. The limit of detection (LOD) was found as 0.5 mM in water and 0.0025 mM in artificial urine. However, the reported study missed specificity and clinical validation study. The SERS-based method has the potential to be developed as a point of care detection test due to small size of the Raman spectrophotometer, albeit it does not show good justification with accuracy or easy-to-use qualitative testing.

#### 3.2.2. Fourier Transform Infrared Spectroscopy (FTIR)

It is empowered to analyze field and mixed opioid samples qualitatively and quantitatively by coupling with chemometric tools (data-driven algorithm to extract information from chemical spectra). FTIR identifies the target molecule based on fingerprint wavelength spectra generated after interaction with a molecule. The extent of light absorbed at each wavelength is identified and calculated, and the outcome is produced as an IR spectrum, which is characteristic of a specific molecular structure. To enhance the signal-to-noise ratio, an interferometer is introduced with IR. It performs a mathematical Fourier transformation to convert the identified signal into a simple interpretable spectrum. The availability of portable IR systems makes this method suitable for onsite drug testing. Some drawbacks are also associated with FTIR, such as strong interference from moisture (H_2_O) in samples, and it is not suitable for an opaque sample. Turner and coworkers [25] have shown the rapid and quantitative discriminative analysis of various opioids from poppy extract with good sensitivity and specificity based on Fourier transform infrared spectroscopy (FTIR). To identify multiple components present in the field sample, authors employed FTIR data with chemometric data analysis to qualitatively and quantitatively define the occurrence of specific opiates. They verified the obtained FTIR data with HPLC-MS data which showed good correlation amongst each other. The calculated LOD for Morphine and Thebaine was 0.13 and 0.3 mg/mL, respectively. The availability of portable IR systems makes this method suitable for onsite drug testing [26].

#### 3.2.3. Absorption Spectroscopy

It emerges as a simple and low-cost, portable analytical tool [27]. It is based on Beer-Lambert law, in which the sample absorbs the characteristic wavelength from the UV-visible spectra, and by using the particular wavelength, characteristic absorption spectra of the sample are obtained. The majority of opioids have UV-visible activity, but it is difficult to distinguish them accurately from a complex biological matrix (blood, urine, urine, etc.) due to interference. Therefore, specific interaction/separation chemistry is required to improve specificity and sensitivity. The gold nanoparticles have strong surface plasmon resonance (SPR) absorptions, with high extinction coefficients in the visible region. Melamine is an electron-rich nitrogen compound that has a high affinity towards gold nanoparticles. Hence, entrancing leverage from the aforementioned properties, the melamine-coated gold nanoparticles protect the nanoparticle from forming aggregates. The aqueous solution of the melamine-coated nanoparticles appears wine-red (λ = 520 nm). In the presence of morphine and codeine, a visible color change (redshift) from wine red to blue is observed (λ = 690 and 675 nm, for morphine and codeine, respectively) due to the aggregation. It can be observed by a UV-visible spectrophotometer or naked eye. This aggregation is triggered by the formation of H-bond between melamine and opioids. The proposed method displayed a linear range of 0.07 to 3 µM for morphine and 0.03 to 0.8 µM for codeine with limits of detection (LOD) of 17 and 9 nM, respectively. Moreover, melamine-coated AuNPs used for the detection of morphine and codeine in biological fluids and pharmaceutical formulations have shown satisfactory results [28]. Mohseni and co-workers have also developed an array tool for opioid analysis. In this approach, they have used four different sizes of gold nanoparticles as an array and generated distinguishing colorimetric responses with the interaction of different members of the opioid family. Based on the colorimetric response from the array, color difference maps were created to provide a visual tool for classifications and semi-quantitative analysis.

Another colorimetric method was developed by Kangas et al. [29] by using eosin Y dye for the detection of fentanyl and fentanyl adulterated with cocaine or hydrocodone. This method can be implicated in a solution or on a solid support system (paper). The dye eosin Y generates different colorimetric responses after reaction with fentanyl (violet), fentanyl+ hydromorphone (magenta), and fentanyl+ cocaine (lilac) at pH 7 in PBS buffer. The authors also printed eosin-Y dye over paper with a regular desktop printer and got a satisfactory discriminatory response between fentanyl and mixed fentanyl. The advantage of this assay includes the use of safe non-toxic reagents, the use of low sample volume, ease of use, and portability. However, the assay needs further validation with real field and biological samples. 

#### 3.2.4. Fluorescence Spectroscopy

It minimizes the limitation of absorbance spectroscopy, based on the principle of transition of the excited-state electron to the ground state followed by emission of a wavelength of light. Shcherbakova and co-workers [30] developed a novel approach for opioid sensing based on the interaction of opioids derivative with the elastic binding pocket of fluorescence derivatives of acyclic cucurbituril (aCBs). These aCB derivatives are covalently linked with naphthalene through different degrees of steric hindrance in their binding pocket, which target different members of the opioid family (i.e., morphine, heroin, and oxycodone). The presence of target opioids (morphine and heroin) leads to alteration in fluorescence emission of the terminal naphthalene-aCB derivative and prompts a turn-on response as a consequence of the enhanced inelasticity. In the case of oxycodone, the fluorescence emission is reduced, as it acts as a fluorescent quencher. The authors have proposed the mechanism that the naphthalene walls and subsequent binding pocket in addition to the guest uptake are driving forces for fluorescence turn-on or turn-off phenomenon. This alteration in fluorescence is correlated with the concentration and type of opioids. The LODs in the spiked buffer were 0.07 ppb for morphine and 82.5 and 2.78 ppb for the morphine metabolites viz. morphine-3-glucuronide (M3G) and normorphine (NMOR), respectively. The LOD in spiked human urine was 27.5, 972, and 71.3 ppb for morphine, M3G, and NMOR, respectively. The availability of a handheld fluorescence spectrophotometer makes this method feasible for POC detection [31].

#### 3.2.5. Interferometer

The interferometry-based approach relies on a change in the refractive index in presence of the analyte, facilitating high sensitivity. An aptamer-based assay has been reported using a compensated interferometric reader (CIR) [32]. The aptamers against various target opioids (oxycodone, noroxycodone, hydrocodone, norhydrocodone, fentanyl, norfentanyl) with binding affinity between 0.66–4.49 nM were used in this study. The CIR offers simultaneous measurement of reference and test samples. It is intended for accurate measurement of the effect of different factors such as surface irregularities and refraction index of solution over a beam of light (electromagnetic wave). A free solution assay (FSA) was developed for the quantitative detection of opioids with CIR, based on opioid-aptamer complex formation and measurement of intrinsic solution-phase properties. In FSA, a sample (spiked urine with opioids) is divided into two parts. One part is mixed with an aptamer developed against respective opioids, which acts as a test sample, and the other part of the sample solution without aptamer acts as a reference solution. The change in the pattern of interference is measured with help of compensated interferometric reader and correlated with opioid concentration. The reported LODs for opioid assay range from approximately 28 to 81 pg/mL (90−245 pM), with a detection range of 0.14–100 nM. This assay (FSA-CIR) showed excellent response in terms of sensitivity, minimal required volume, and operation cost compared to other available techniques or kits such as LC/MS, ELISA, and Mayo clinic kit.

#### 3.2.6. Chromatography

Chromatography is another technique that is extensively used for opioid detection. It hinges on the principle of differential distribution of sample between mobile and stationary phase and separates the sample into individual spots or peaks. High-pressure liquid chromatography (HPLC) and gas chromatography (GC) are commonly used methods in the pharmaceutical, law enforcement, and health care sectors. It separates the different compounds from the complex sample and later identifies and quantifies them based on property of individual components [33]. The chromatography system is empowered to analyze opioids with high specificity and sensitivity by coupling it with diode array [34], absorbance or fluorescence spectroscopy [35,36,37], and mass spectrometry (MS) [38]. GC-MS is much more sensitive than HPLC-UV but not as sensitive as HPLC-MS. [39,40].

Other techniques that are sometimes used for opioid detection include capillary electrophoresis [41], and X-ray diffractometry [42]. Since the turn of the century, sophisticated and automated methods have been developed for detecting opioids in complex biological samples (mixed samples), with an emphasis on POC testing to replace RDTs with quantitative methods.

### 3.3. Advance Point of Care (POC) Techniques for Opioids Testing

POC opioid detection can be significantly enhanced with electrochemical detection since it can easily be integrated with miniaturized readout circuitry [43]. With the use of carbon nanotube and double-stranded DNA functionalized electrodes in which a specific DNA sequence identifies opioids of interest, differential pulse voltammetry has recently been used for opioid detection. Based on its high sensitivity, specificity, and reproducibility, this platform is ideal for detecting opioids in biofluids (e.g., urine, plasma) [44]. Recent studies have reported an aptamer sensor for opioid detection based on gold nanoparticles and iron oxide nanoparticles. The aptamer switches its conformation when in contact with the target opioid and inhibits the flow of electrons through the modified electrode [45]. Consequently, the opioid concentration directly correlates with the current response. With a sensitivity of pM range, this biosensor can detect opioids in biological fluids [23]. Sensors based on electrochemical technology have the advantage of being readily miniaturized and integrated into wearable devices. A continuous opioid monitoring system based on electrochemical sensors has also been developed [46]. The Bio-Mote is a wireless, integrated miniature sensor (250 × 770 µm^2^) that can be injected into interstitial fluid just under the skin using a 16-gauge syringe. Wirelessly powered, the Bio-Mote sends measurements back to the wearable device placed above it (e.g., a smartwatch or patch). A wearable micro needle-based sensing patch was developed for continuous monitoring of fentanyl levels in the interstitial fluid using a similar approach [47]. The fentanyl undergoes irreversible electrochemical oxidation due to the presence of a tertiary amine group corresponding to the formation of quinone imine (norfentanyl) at the carbon electrode. Using a skin-like phantom gel, this sensor can monitor the presence of fentanyl up to nanomolar levels. The integrated electrochemical sensors measure in vivo opioid levels in real-time closing the treatment loop. A system like this could be extremely useful for enhancing the detection of OIRD and for opioid use treatment programs.

There is great demand and need for law enforcement/forensic laboratories/healthcare providers and PWUD for the development of a specific, sensitive, and accurate analytical approach for the separation and quantification of different members of the opioid family. Fentanyl is an extremely potent synthetic opioid that is prone to abuse, so it is imperative that it can be detected rapidly and on-site. To fulfill the unmet need, a “Lab-on-a-Glove” was reported as a method of detecting liquid or powdered fentanyl with a detection range of 10–100 µM and a measurement time of under 1 min [48]. Moreover, in the sensor, there was no interference from common street drug adulterants (e.g., caffeine, sugar, acetaminophen, etc.) that can compromise detection or result leading to false positives. By seamlessly integrating opioid detection into their existing workflow, this technology enables first responders, medical professionals, and field agents to detect opioids quantitatively. The ability to recognize the presence of opioids enables first responders to take the necessary precautions while administering the right medication to provide rapid treatment along with protecting themselves and others. This can be used to alert PWUD of unexpected additives (e.g., fentanyl content and amount) as part of a harm reduction strategy. With this wearable-based “swipe, scan, sense, and alert” strategy, chemical analytics are at users’ fingertips. Technology such as this can be extended to substance abuse treatment programs that currently rely on self-reporting and random drug testing.

An overview of recent advances in opioid sensing is shown in Table 1. As can be seen from the table, GC/MS provides excellent results (sensitivity and specificity) compared to other opioid detection technologies. It is difficult, however, to envision GC/MS as a tool outside of a centralized testing facility due to its high running costs, operational complexity, and difficulty in miniaturizing. It is evident from the available literature that electrochemical techniques are the most reliable and successful solution as they offer rapid, sensitive, and specific opioid detection at a low cost. It may be possible to convert the electrochemical technique into a portable device suitable for street use and POC settings with the expected advancement of microelectronics and electrochemical detection of opioids.

## 4. Established Devices Used for Detection of OIRD: Requiring Modification to Impact the Public Health

While new technologies are being developed and miniaturized to detect opioids in vivo, there remains a need to detect the downstream effects of opioid excess, such as opioid-induced respiratory depression. Although there is some variability in definitions of respiratory depression (based on studies), at its most basic level, respiratory depression is the result of decreased minute ventilation, a combination of tidal volume and respiratory rate. Diagnostics of OIRD present a unique set of challenges. The respiratory status of patients in non-continuously monitored environments is often assessed using infrequent vital signs such as respiratory rate and oxygen saturation. Despite their value, these measurements serve only as surrogate indicators of an individual’s respiratory health. A marked decrease in ventilation leads to hypercarbic respiratory failure, the sine qua non of severe opioid toxicity [61]. In many clinical settings, respiratory rate and oxygen saturation are often used to assess respiratory status, with sedation assessment and capnography also playing a role, particularly in inpatient settings. As a result, technologies used to detect OIRD, particularly those that can be scaled and conceivably used for out-of-hospital public health interventions, commonly measure one aspect of decreased ventilation (often respiratory rate) or a surrogate measure (e.g., decreased oxygen saturation). In the absence of low-cost, scalable modalities to measure minute ventilation, surrogate measurements, balancing their sensitivities and specificities, may be needed from a public health perspective. Early detection and intervention with naloxone or supportive respiratory care are highly reliable ways to reverse OIRD and prevent irreversible morbidity or mortality. It is important to modify existing technologies and develop new, innovative detection modalities to detect respiratory depression faster to quickly identify OIRD and connect victims to therapies.

### 4.1. Pulse Oximeter

Pulse oximeters are used for non-invasive, inexpensive measurements of oxygen saturation, heart rate, as well as respiratory rate (only in advanced models). By measuring the variation in absorption of light in oxygenated and deoxygenated blood, and extrapolating SpO_2_ based on a reference standard, oxygenation can be calculated. Although with advancement in technology the pulse oximeter performance has been improved to function under adverse conditions like excessive motion and low perfusion, yet the issue of false alarms remains elusive due to a variety of factors and can contribute to provider fatigue [62]. Primarily pulse oximetry is a measure of oxygenation, not ventilation. Hence, decreased ventilation can be present for a longer time before being detected by a pulse oximeter, and the use of supplemental oxygen can further delay this response. In cases of acute opioid overdose, it could be asserted that desaturation being a late sign of the toxicity event, could further taper the chances of therapeutic intervention for an already emergent condition. However, from a public health perspective, in the existing opioid overdose epidemic, the reliability, low cost, high clinical sensitivity for overdose in opioid use contexts, pulse oximetry is a compelling technology. OIRD leading to hypoventilation or decrease in oxygen saturation level causes a rapid drop in oxygen at opioid overdose. The rise of hypoventilation upon overdose increases the sensitivity of pulse oximeter in terms of SpO_2_ [63]. For pulse oximetry to have a scalable public health intervention impact, it is imperative to couple them with devices capable of summoning help (e.g., friends, family, bystanders, EMS personnel) upon detection of a sustained critical desaturation event. The drive to incorporate pulse oximetry on a variety of mobile phone-connected wearables (e.g., watches and bracelets), permits monitoring outside of hospital environments, making the barrier to use and decreased chance for stigma favorable.

### 4.2. Capnography

Capnography, although is less commonly used than pulse oximetry, is more sensitive in detecting OIRD as it is capable of indirectly measuring ventilation from exhaled CO_2_ using an infrared sensor. When plotted as a function of time, these values provide a more accurate measure of ventilation and can detect OIRD earlier than a pulse oximeter. OIRD can be detected using capnography on basis of the observed decline in end-tidal CO_2_ (ETCO_2_) peek detection (indicating reduced respiratory rate) and/or a higher concentration of CO_2_ (indicating retention). Respiratory rates that fall below predetermined “normal” values or ETCO_2_ levels above programmed alarm cutoffs would trigger a notification. These alarm set values may need to be adjusted as per an individual’s normal physiology due to several factors such as medical comorbidities. Capnography also suffers from some disadvantages such as ETCO_2_ does not always readily correlate to CO_2_ levels in the blood. These values are differentiated by a gradient that is a surrogate to physiologic dead space and can be variable [64]. Further, capnography requires the use of specialized equipment to quantify exhaled breath and requires the patient to wear a sampling line (e.g., nasal cannula) appropriately, which can be challenging and uncomfortable [65]. Finally, from a public health point of view, the cost involved in using a capnography system prevents it from being used as a technology that could scale and be deployed to mitigate risk from opioid overdose in the community. Hence, there must be a low-cost, less invasive variation of capnography, connecting victims to help in the cases of hyper-carbic respiratory failure.

## 5. Novel, Contactless Sensing Modalities: Sonar, Radar, Computer Vision

Contactless monitoring systems provide a fascinating method of monitoring for OIRD as they do not require the user to instrument themselves or wear anything, and the technologies to transmit and receive these signals are in some cases ubiquitous. One family of systems involves using active sonar with commodity devices such as smartphones and smart speakers [66]. These systems leverage the internal hardware of these devices (e.g., speaker, microphones, processor) and thus can be done with the software only. These systems use frequency-modulated continuous waveform (FMCW) and convert the device’s speaker and microphone into a short-range active sonar system. At specific moments of high risk (e.g., before opioid self-injection or during sleep) the devices transmit custom, inaudible, FMCW where the transmitted frequency increases linearly with time between 18 and 22 kHz (wavelength 1.7–1.9 cm) within a duration of 10 ms [67,68]. The custom acoustic signals reflect off a surface (in this case, a moving chest during respiration), and the echo arrives back to the device’s microphones after a time delay. The echo is captured by the microphones, distance from the reflector is processed, and a respiratory rate is generated [67]. Experimental systems using these techniques developed by researchers have shown high accuracy for measuring respiratory rate and good sensitivity for identifying breathing patterns associated with opioid toxicity (>87% for RR < 8 breaths/minute) including prolonged apnea (>95%) [67]. These systems could be useful from a harm reduction standpoint, whereby a person who uses drugs could monitor their breathing when engaging in a high-risk opioid use event (e.g., self-injection), where there is a circumscribed window of elevated risk. For people using chronic opioid therapy to manage a pain condition, nocturnal monitoring could be done as this represents a higher risk environment, particularly if the patient is prescribed other central nervous system depressants (e.g., benzodiazepines), has obstructive breathing or consumes alcohol. The generally low cost and ubiquity of smartphones and smart speakers make active sonar a novel and compelling OIRD solution from a public health standpoint, as does the fact that these devices have built-in connectivity to summon help should an emergency arise.

Radar-based systems are also examples of contactless systems. Radar-based systems transmit electromagnetic waves into the environment, while reflected waves from the thoracic cavity are captured at the transceiver and used to determine a patient’s respiratory rate. This method of monitoring has its challenges. Due to radar’s sensitivity to the environment and its high travel speed, it can be difficult to distinguish the intended target (a patient’s variably shaped thoracic wall) from other potential sources of movement. In addition to their high sensitivity in detecting the millimeter movements involved in respiration, radar-based systems are also highly sensitive to noise introduced by non-respiratory body movements. Since these devices use electromagnetic waves, a variety of factors (for example physical movements, metal jewelry, and radio-frequency interference) need to be considered, which may affect the practical applicability at the point of need (PON) [69].

Contactless respiration measurement is also possible through computer vision with remote system settings [70]. The respiratory rate of an individual is calculated (based on movement or pixel intensity variation) by extracting thoracoabdominal pixel changes and tracking them from sequential video frames. One system by Chaterjee et al. using a consumer-grade camera observed high correlation (*r* = 0.88) with ground truth and 93% of measurements within 3 breaths/minute [71]. A computer vision-based system can utilize existing commercially available cameras, which are inexpensive and require little additional technical knowledge on the part of the user. Photoplethysmography, another computer vision-based technique, can be combined with this contactless monitoring technique to measure heart rate based on subtle changes in coloration in the face (depending on variations in blood flow) that are imperceptible to the human eye. Vital sign information combined with respiratory data could be useful for identifying individuals who may be experiencing an opioid overdose. McDuff and colleagues have demonstrated that using commodity hardware computer vision techniques, heart rate can be identified to within four beats/minute [72]. Computer-vision-based systems have several limitations in addition to privacy concerns, such as their sensitivity to low light levels, increased interference with magnified images, and errors caused by camera movement. A drawback of contactless systems, such as other devices, includes the need for subjects to be relatively stationary when measurements are taken, otherwise gross motor movements can overpower the signal associated with tidal breathing. An overview of recent advances in detection and diagnosis of OIRD is shown in Table 2.

## 6. Conclusions

Although ventilatory impairment is a well-understood side effect of opioid administration in a hospital setting, the current opioid epidemic brings to the forefront an urgent need for the development of additional accessible means of detection outside of the hospital. New technologies are required for not only monitoring the signs and symptoms of OIRD but also to alert first responders through integration with mobile devices. In the early detection and treatment of chronic opioid addiction, new monitors could allow for more rapid intervention. Systems designed to identify in vivo opioids in real-time could be used to significantly improve the monitoring of chronic opioid (and abstinence) therapy. It will, however, be critical to evaluate and test devices extensively to establish the safety of these devices. In addition to monitoring advancements, limitations such as alarm fatigue will need to be addressed and the utilization of public resources (e.g., 911/EMS systems, drug monitoring programs) will need to be taken into account when helping persons who have overdosed on opioids. The outcome of this review suggests there is a critical need for remote respiration monitoring and POC testing techniques, which can offer excellent sensing and detection parameters for OIRD and opioid use. Improved diagnosis and detection technologies can provide critical information both to those who are at risk and those who treat OIRD. Public health can be significantly impacted by optimizing these technologies to combat the opioid crisis.

## Figures and Tables

**Figure 1 biomedicines-10-00743-f001:**
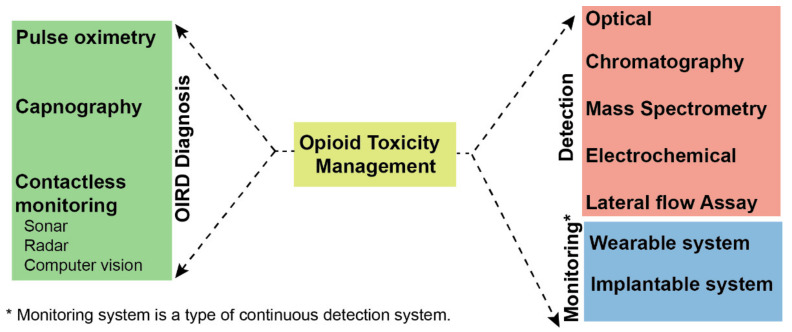
Overview of techniques for management and detection of opioid toxicity.

**Figure 2 biomedicines-10-00743-f002:**
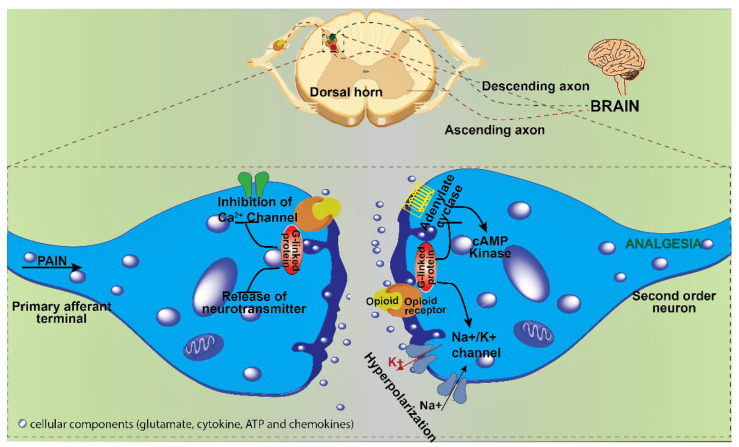
Mode of action of opioids in pain suppression at the cellular level. The binding of opioids to opioid receptors inhibits the calcium channel and blocks the release of neurotransmitters (represented as silver blobs in synaptic space, which get depleted upon binding of opioids) in the presynaptic neuron. In the postsynaptic neuron, the binding of opioids leads to hyperpolarization, preventing the transmission of the signal of pain and hence analgesia. (The image is illustrated with software chembio draw v.19 and Adobe Illustrator CS6).

**Figure 3 biomedicines-10-00743-f003:**
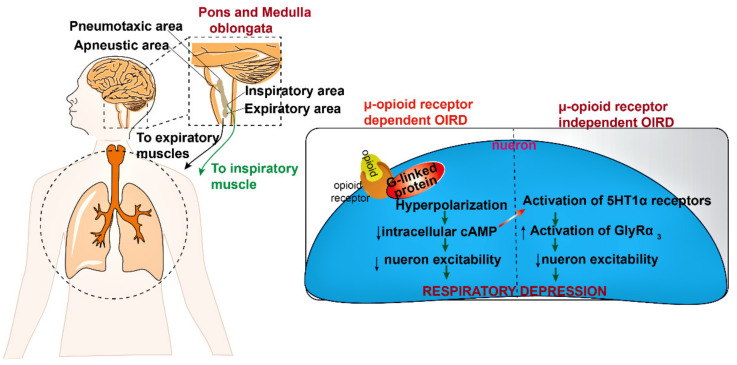
The pathophysiology and molecular mechanism of opioid-induced respiratory depression. Opioids activate the G-protein coupled μ-receptor and hyperpolarize the neurons through Na^+^/K^+^ channels and decrease the intracellular cAMP level. This alteration in cAMP levels further activates the 5HT1α and GlyRα_3_ receptors which lead to a decrease in the neuron excitability and depress breathing by inhibiting the respiratory center (pons and medulla). (5-HT-serotonin; GlyR3-glycine receptor type-3, cAMP-cyclic adenosine monophosphate. Symbols: upward arrow-increased effect; downward arrow-decreased effect). (The image is illustrated with software chembio draw v.19 and Adobe Illustrator CS6.).

**Table 1 biomedicines-10-00743-t001:** Summary of opioid detection techniques.

	RDT [17]	FTIR [25,49,50]	Raman [24,51]	GC-MS [22,52,53]	LC-MS [20,54]	HPLC [37,55,56]	Interferometry [32,57]	Electrochemical [43,58,59]
**Detection Principle**	Immuno chromatography	Identification of fingerprint spectra	Interaction of Light with molecule	Separation by vapor pressure and distribution constant and Identification by *m*/*z*	Separation of mixture based on chemical/physical properties and identification by *m*/*z*	Separation based on Distribution and Identification with UV spectroscopy	Measurement of intrinsic solution phase properties	Voltammetry or Amperometry
**Specimen**	Urine/Saliva	Blood/Powder	Urine	Urine/Saliva	Urine/Blood/saliva	Plasma/Urine	Urine	Blood/Urine/Saliva
**Sensitivity**	Moderate	Moderate	Moderate	Excellent	Excellent	Moderate	Excellent	Excellent
**Specificity**	Excellent	Moderate	Moderate	Excellent	Excellent	Poor	Good	Good
**LOD**	µM	µM	mM range	pM	pM	nM	pM	pM
**Response Time**	5 min	1 min	5 min	30 min	<30 min	30 min	60 min	1 min
**Skillset**	Low	Moderate	Moderate	High	High	High	Moderate	Low
**Use case**	POC	POC/Lab	POC/Lab	Lab	Lab	Lab	Lab	POC/Lab
**User ***	LE/HC	LE/HC	LE/HC	HC	LE/HE	HC	HC	LE/HC
**Cost/test** **[Instrument price]**	Very low (Approximately $1–5)	Moderate, $50–100 ^#^ ($100,000 + new, advanced models Portable: $10,000–60,000)	Moderate, $80–120 ^#^ ($100,000 + new, advanced models Portable: $10,000–60,000)	High $100–200 ^#^ ($200,000 + new, advanced models)	High $100–200 ^#^ ($200,000 + new, advanced models)	Low $25–50 ^#^ ($80,000 + new advanced models)	Moderate $50–80 ^#^ ($50,000 + new advanced models)	Very low (NA) ($6000 + new, advanced models Portable: $500–5000)

* LE = Law enforcement at the field; HC = Health care, ^#^ cost/test mentioned here is for comparison purposes based on standard sample analysis [60].

**Table 2 biomedicines-10-00743-t002:** Summary of devices used in the detection and diagnosis of OIRD. (HR = heart rate; RR = respiratory rate; EtCO_2_ = end-tidal CO_2_; SpO_2_ = oxygen saturating point).

	Pulse Oximetry	Capnography	Contactless Systems	Remote Systems
**Detection principle**	SpO_2_, HR	EtCO_2_, RR	RR	SpO_2_, HR
**Sensitivity**	High	High	Moderate	Moderate-High
**Specificity**	Moderate to High	High	Moderate	Moderate-High
**Response Time**	Slow	Fast	Medium	Medium-Fast
**Skillset**	Low	Low-Moderate	Low-High	Low-Moderate
**Cost**	Low	Moderate	Low-High	Low-Moderate

## Data Availability

Not applicable.

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
