# Peer review of "Current and Future Perspective of Devices and Diagnostics for Opioid and OIRD"

_biomedicines, 2022, doi:10.3390/biomedicines10040743_

Round 1

Reviewer 1 Report

I believe the article is very interesting and well written. 

Author Response

Reviwer-1

I believe the article is very interesting and well written. 

Response: Thank you so much for evaluating our manuscript. We are thankful to the reviewer for appreciating our work and positive recommendations of manuscript for publication.

Reviewer 2 Report

The submitted manuscript „Current and future prospective of Devices and Diagnostics for  OIRD” presents an important problem of current medical care – continuous monitoring of health conditions.

In my opinion, there are at least three levels of detection of opioids and their consequences in the manuscript: detection of drugs as substances, detection of opioids in the biological samples (samples of fluids or implanted sensors) – and the detection of respiratory problems (a broad subject, as such problems are caused by various health issues, not only opioid use).
The title of the paper does not represent the presented topics. It could be rather something closer to “for opioids and OIRD”

The description of opioids and their biological activity is detailed and quite advanced, and similar level of presentation may be expected in other topics.

Unfortunately, the Clinical detection of opioids was prepared, in my opinion, without deeper study of many described methods.

First of all, the authors should specify the character of samples analyzed (blood, urine, saliva, drugs, plant extracts) and sample preparation procedure (direct analysis, extraction etc.), especially when discussing method specificity and sensitivity. Mixing the length of analysis with the time needed for sample delivery cannot be accepted without discussion of on-site analysis devices (included in the other parts of text).

Including chromatography in the part of Optical Detection techniques is also misconception – it is a separation technique (with much better resolution than suggested in the text, the phrase “poor ability to differentiate among compounds” is just not true with modern columns, the use of a 1999 reference is not a good idea). Liquid chromatography is sometimes coupled with spectroscopic (UV, FL) detection, recently rather with MS detection for better sensitivity and identification.  Gas chromatography is practically not used with spectroscopic detection (FID got different background).

The last part of 3.1 paragraph is the example of sample issue (drug versus biological sample): “In particular, mass spectrometry has outstanding specificity for heterogenous samples; however, because of the ionization method and large magnetic field needed for operation, this technique is extremely challenging to miniaturize. However, DE Tector flex and RoadRunner, portable MS devices used by law enforcement to detect opioids and derivatives with nanogram sensitivity, are two such devices available today [20].

In the case of Electrochemical detection, the description is broader and includes critical details. However, aptamer sensors or aptamer selection is also used in methods described in part 3.1 (it could be classified as a component of sample preparation as selective separation), and it is responsible for the use of such methods (specific electrodes).

Then there is a “Lab-on-a-Glove” and a very interesting fragment on detection of opioids during intervention – as powders, drugs – as compared to opioids already in the body of affected person. It is critical in the case of first responders - and it calls for a better introduction to manuscript topics.

Table 1 – the LOD comparison cannot be accepted if different character of samples affects the analysis. HPLC is actually HPLC UV, whereas LC-MS gives completely different effect. Does interferometry without prior aptamer use provide specificity?
The cost/test is an excellent idea as long as a cost PER  TEST is calculated in the case of instrumental systems, to compare with RDT.

In the description of Pulse oximeters there is a sentence: “However, from a public health perspective, in existing opioid overdose epidemic, the reliability, low cost, high clinical sensitivity for overdose in opioid use contexts, pulse oximetry is a compelling technology.” Please explain the fragment “high clinical sensitivity for overdose in opioid use contexts” for parameters monitored by pulse oximeters.

The contactless devices are really interesting, however, are radar-based systems safe in the case of metal jewelry and clothing elements? There is no reference included for this part and no information of wavelengths.

What is a difference between contactless and remote systems (Table 2)?

Other issues:

There are examples of endogenous opioids, whereas no examples of exogenous opioids are mentioned at the beginning of chapter 2. This may be not obvious for some readers. Enkephalins etc. are short peptides and calling them polypeptides is incorrect. The term “nociceptor” is quite interesting and the explanation for the kind of sensation detected may be helpful.

Figure 1: For detection methods: it is mass spectrometry, not spectroscopy, and there is a spelling error in Electrochemical. Amperometry is the correct form.
Implantable – should it be at the same level as Voltammetry and Amperometry?
Some methods mentioned in the figure are not presented in the text.
The OIRD methods are really used in management and detection of opioid TOXICITY consequences, whereas the detection methods, as described in the text, rather indicate opioid presence?

Some explanation for parts of Figure 2 is needed, including silver orbs floating between synapse parts.

Please verify “ prospective” in the title.

Please use “in vivo” in italics consequently in the text.

Please correct H2O and CO2 to the proper formulas with indices.

Is there a difference between RDTs and RDTS?

Table 1: Immuno chro-mate—ography?

Please consider the positions of commas in the sentence (sensors, and detection) ” In addition to respiratory monitoring devices such as pulse oximetry, capnography, and contactless monitoring systems, novel implantable sensors, and detection systems such as optical detection and electrochemical detection techniques are being developed to identify the presence of opioids both in vivo and within the environment.”

Please check the sentences for style and repetitions:

 “also called as MOR, DOR, KOR and respectively, as per current standard terminology”

“The binding of opioid to its receptors leads conformational changes in the receptor leading to activation of G coupled protein”

There are abbreviations without the full version (PWUD, POC, MALDI)

Author Response

Revewer-2

We thank this referee for their critical evaluation of our manuscript. Below you will find a point-by-point reply to all comments that have been made. We are submitting the track-change version of previous version as well as clear version, so that reviewer can compare/observe the broad level changes.

Point 1: The submitted manuscript „Current and future prospective of Devices and Diagnostics for OIRD” presents an important problem of current medical care – continuous monitoring of health conditions. In my opinion, there are at least three levels of detection of opioids and their consequences in the manuscript: detection of drugs as substances, detection of opioids in the biological samples (samples of fluids or implanted sensors) – and the detection of respiratory problems (a broad subject, as such problems are caused by various health issues, not only opioid use). The title of the paper does not represent the presented topics. It could be rather something closer to “for opioids and OIRD”

Response: We appreciate reviewer constructive suggestions, now we have improvised the manuscript title “Current and Future Perspective of Devices and Diagnostics for Opioid and OIRD”.

Point 2: The description of opioids and their biological activity is detailed, and quite advanced, and similar level of presentation may be expected in other topics. Unfortunately, the Clinical detection of opioids was prepared, in my opinion, without deeper study of many described methods.

Response: Thank you for constructive suggestion, now we have improvised and enriched the opioid detection section and highlighted in yellow for kind review. We are submitting the track-change version of previous version as well as clear version, so that reviewer can compare/observe the broad level changes.

Point 3: First of all, the authors should specify the character of samples analyzed (blood, urine, saliva, drugs, plant extracts) and sample preparation procedure (direct analysis, extraction etc.), especially when discussing method specificity and sensitivity. Mixing the length of analysis with the time needed for sample delivery cannot be accepted without discussion of on-site analysis devices (included in the other parts of text).

Response: Yes, this is a good point. The manuscript has been revised based on reviewer suggestion for selecting specimens and clinical testing using conventional laboratory techniques. To facilitate the review, process the text that has been added is given below:

Conventional laboratories techniques for opioids testing

i). Raman Spectroscopy

ii). Fourier Transform Infrared Spectroscopy (FTIR)

iii). Absorption spectroscopy

iv). Fluorescence Spectroscopy

v) Interferometer

vi). Chromatography

Point 4: The last part of 3.1 paragraph is the example of sample issue (drug versus biological sample): “In particular, mass spectrometry has outstanding specificity for heterogenous samples; however, because of the ionization method and large magnetic field needed for operation, this technique is extremely challenging to miniaturize. However, DE Tector flex and RoadRunner, portable MS devices used by law enforcement to detect opioids and derivatives with nanogram sensitivity, are two such devices available today [20].

Response: We appreciate the reviewer's concern. Now we have enriched the manuscript with the type of biological specimens used for opioid detection with their pros and cons. It is possible to define additional preparative steps or procedures based on the "skillset" for each test type. As an example, sample analysis with a "low skillset" is associated with direct sample analysis without any pre-partitive step (Table-1).

We agree with the reviewer's concern about the sensitivity and specificity, that various factors can affect the sensor characteristics of a particular “technique" by varying a sensing probe, matrix, mediator complex, analysis time, and transducer system. Hence it will very difficult to justify the preference parameter of techniques. However an attempt was made to normalize the comparison, made in table -1 based on the available literature and Drug checking- evidence review annual report published by the British Columbia center of substance abuse (https://www.bccsu.ca/wp-content/uploads/2017/12/Drug-Checking-Evidence-Review-Report.pdf).  

Point 5: Including chromatography in the part of Optical Detection techniques is also misconception – it is a separation technique (with much better resolution than suggested in the text, the phrase “poor ability to differentiate among compounds” is just not true with modern columns, the use of a 1999 reference is not a good idea). Liquid chromatography is sometimes coupled with spectroscopic (UV, FL) detection, recently rather with MS detection for better sensitivity and identification.  Gas chromatography is practically not used with spectroscopic detection (FID got different background).

Response: We agree and appreciate the reviewer valuable suggestions, now we have reorganized the detection techniques into current clinical, conventional laboratories and advance point of care categories. Now we have substituted the statements in regards of HPLC and updated the reference.  

Point 6: In the case of electrochemical detection, the description is broader and includes critical details. However, aptamer sensors or aptamer selection is also used in methods described in part 3.1 (it could be classified as a component of sample preparation as selective separation), and it is responsible for the use of such methods (specific electrodes). Then there is a “Lab-on-a-Glove” and a very interesting fragment on detection of opioids during intervention – as powders, drugs – as compared to opioids already in the body of affected person. It is critical in the case of first responders - and it calls for a better introduction to manuscript topics.

Response: We appreciate the reviewer concern, now we have removed the “aptamer selection” statement from the manuscript. Now we have improvised the “Lab on gloves” and added a dedicated paragraph with proper introduction and possible application of sensor.

Point 7: Table 1 – the LOD comparison cannot be accepted if different character of samples affects the analysis. HPLC is actually HPLC UV, whereas LC-MS gives completely different effect. Does interferometry without prior aptamer use provide specificity?

Response: Please see our response in point 3. We have added an additional row in table-1, with type of sample used for analysis.

In the original manuscript, Kammer et al., 2019 haven’t conducted such experiments. However same techniques with similar transducer principle “lacking the desired specificity in the absence of suitable probe (Synthetic- nanoparticles, chemicals, MIP; Biological- aptamer, antibody, affirmers and enzymes).

Point 8: The cost/test is an excellent idea as long as a cost PER TEST is calculated in the case of instrumental systems, to compare with RDT.

Response: We appreciate revisers valuable suggestion, now we have added the cost/ test in the table 1.

Point 9: In the description of Pulse oximeters there is a sentence: “However, from a public health perspective, in existing opioid overdose epidemic, the reliability, low cost, high clinical sensitivity for overdose in opioid use contexts, pulse oximetry is a compelling technology.” Please explain the fragment “high clinical sensitivity for overdose in opioid use contexts” for parameters monitored by pulse oximeters.

Response: We appreciate the reviewer concern, now we have explained the indicated fragment in the manuscript.

Point 10: The contactless devices are really interesting, however, are radar-based systems safe in the case of metal jewelry and clothing elements? There is no reference included for this part and no information of wavelengths.

Response: We appreciate the reviewer concern, now we have added the desired information’s in the manuscript along with the references.

Point 11: What is a difference between contactless and remote systems (Table 2)?

Response: Remote-based systems are also examples of contactless systems.  Which is primarily based on video technology and machine vision technology. It offers the measurement of heartbeat and respiration rate remotely to heath care providers remotely with computer assisted technology. Vital Sync™, OxiNet III system are few examples of commercially available remote based system (https://doi.org/10.1371/journal.pone.0071384).

Table 2 belong to the section five of the manuscript. We have mentioned there (in first paragraph line 448-452) while remote sensing is the measurements of heart and respiration rates through Telemedicine which requires sonar, radar, and computer vision (Line no 488).

Point 12: Other issues: Minor

  1. There are examples of endogenous opioids, whereas no examples of exogenous opioids are mentioned at the beginning of chapter 2. This may be not obvious for some readers. Enkephalins etc. are short peptides and calling them polypeptides is incorrect. The term “nociceptor” is quite interesting and the explanation for the kind of sensation detected may be helpful.

  1. Figure 1: For detection methods: it is mass spectrometry, not spectroscopy, and there is a spelling error in Electrochemical. Amperometry is the correct form.
    Implantable – should it be at the same level as Voltammetry and Amperometry?
    Some methods mentioned in the figure are not presented in the text.
    The OIRD methods are really used in management and detection of opioid TOXICITY consequences, whereas the detection methods, as described in the text, rather indicate opioid presence?

Response: We appreciate the reviewer suggestion, now we have corrected the typo errors and figure 1.The qualitative and quantitative detection (or opioid presence) method help to healthcare service provider to get the objective data that can be clearly communicated through statistics and numbers. It helps to attain greater knowledge with deep understanding that support in the management of the clinical situation. It also supports into observe emergency situations or events that affect person/ people health. Hence opioid detection has direct role in management of opioid induced toxicity.  As we have tried to justify with our figure 1 in the manuscript.

  1. Some explanation for parts of Figure 2 is needed, including silver orbs floating between synapse parts.

  1. Please verify “ prospective” in the title. (Replace to perspective)

  1. Please use “in vivo” in italics consequently in the text. Done

  1. Please correct H2O and CO2 to the proper formulas with indices. Taken care

  1. Is there a difference between RDTs and RDTS?- No

  1. Table 1: Immuno chro-mate—ography? Corrected

  1. Please consider the positions of commas in the sentence (sensors, and detection) ” In addition to respiratory monitoring devices such as pulse oximetry, capnography, and contactless monitoring systems, novel implantable sensors, and detection systems such as optical detection and electrochemical detection techniques are being developed to identify the presence of opioids both in vivo and within the environment.”
  2. Please check the sentences for style and repetitions: “also called as MOR, DOR, KOR and respectively, as per current standard terminology”

Response: We are grateful for these suggestions. All have been accepted and incorporated

“The binding of opioid to its receptors leads conformational changes in the receptor leading to activation of G coupled protein” There are abbreviations without the full version (PWUD, POC, MALDI)

Response: We appreciate that reviewer has pointed out and checked these minute errors. We have made the changes in the revised version. We have sincerely, read the manuscript and taken care of all the punctuation and grammar mistakes with more diligently and care. Thank you so much to make our paper much better.

Reviewer 3 Report

This is an interesting mini-review on development of new technologies for monitoring of opioid-induced respiratory depression. The Introduction contains some relevant information concerning basic mechanisms of opioid action on nociception and respiratory function. The problem of opioid toxicity as a significant public health concern has been only briefly described, so,  this part  could be expanded by referring to some new epidemiologic data. In the next part of this article,  the main advantages and disadvantages of various  respiratory monitoring devices, also those in development, have been competently discussed. Collectively, this is a well-written paper on an important topic with a strong  emphasis on analytical aspects of opioid pharmacotherapy and intoxication.

Specific remarks:

  1. Introduction

The sentence “Compounds that bind to opiate receptors are referred to as opioids.”  Should be: to opioid receptors.

The sentence “Methadone, fentanyl, and propoxyphene are synthetic analogs of opioids.” These compounds are synthetic opioids, not analogs of synthetic opioids. The term “opioids” refers to all natural, semisynthetic, and synthetic opioids.

2. Fig. 1  Spelling:  comuputer vision

3. Figure 2 well illustrates cellular mode of action of opioids in pain suppression. However, taking into account the topic of this review, a similar  Figure showing mode of action of opioids in respiratory depression (OIRD) would be desired.

Author Response

Reviewer-3

We are thankful to the reviewer for his/her constructive recommendations and for suggesting the revision of the paper.

Point 1: This is an interesting mini-review on development of new technologies for monitoring of opioid-induced respiratory depression. The Introduction contains some relevant information concerning basic mechanisms of opioid action on nociception and respiratory function. The problem of opioid toxicity as a significant public health concern has been only briefly described, so,  this part  could be expanded by referring to some new epidemiologic data.

Response: Thank you for evaluation our efforts. As per the suggestion we have change in the introductory section of the manuscript. Please check under the section 2. Mode of action of opioids in pain suppression and OIRD.

Point 2: In the next part of this article, the main advantages, and disadvantages of various respiratory monitoring devices, also those in development, have been competently discussed. Collectively, this is a well-written paper on an important topic with a strong  emphasis on analytical aspects of opioid pharmacotherapy and intoxication.

Response: We are thankful to the reviewer for appreciating our work and positive recommendations of manuscript for publication.

Point 3: Specific remarks:

  1. Introduction: The sentence “Compounds that bind to opiate receptors are referred to as opioids.”  Should be to opioid receptors.
  2. The sentence “Methadone, fentanyl, and propoxyphene are synthetic analogs of opioids.” These compounds are synthetic opioids, not analogs of synthetic opioids. The term “opioids” refers to all natural, semisynthetic, and synthetic opioids.
  1. Fig. 1 Spelling:  comuputer vision
  2. Figure 2 well illustrates cellular mode of action of opioids in pain suppression. However, taking into account the topic of this review, a similar  Figure showing mode of action of opioids in respiratory depression (OIRD) would be desired.

Response: We have sincerely, read the manuscript and taken care of all the mistakes with more diligently and care. We are very thankful for reviewer has checked these minute errors.

Round 2

Reviewer 2 Report

I would like to thank the Authors for their determination in improving the text. The detailed answers to issues indicated in the review, as well as the changes in the text indicate a remarkable positive change in the manuscript.

The presented information is now well organized and coordinated.

I am sorry that the Authors did not include LC-MS data in Table 1, as the sensitivity is usually impressive, although the number of columns is already high and it may be difficult to accomodate one more into the table. The information is included in the text.

Additional issue:

In the text, the problem of color change (absorption spectroscopy, line 238 and following) has to be verified, the indicated wavelengths were used to distinguish morphine and codeine, the general color change was from red to blue (520/690).

However, the modifications suffer from the limited time the Authors had available to provide so substantial text changes.

The first version of the text was prepared practically without noticeable language errors. The current version must be corrected (in the parts that were modified) as there are numerous errors (grammar) and in some cases the units of measure were affected (for example lines 243-244).

The number of errors is rather high and I marked some of them in the text instead of preparing the list.

Author Response

I would like to thank the Authors for their determination in improving the text. The detailed answers to issues indicated in the review, as well as the changes in the text indicate a remarkable positive change in the manuscript. The presented information is now well organized and coordinated.

Response: Thank you for acknowledging our efforts.

I am sorry that the Authors did not include LC-MS data in Table 1, as the sensitivity is usually impressive, although the number of columns is already high and it may be difficult to accomodate one more into the table. The information is included in the text.

Response: We have incorporated the LC-MS data in the Table1 Column No. 6.

Additional issue:

In the text, the problem of color change (absorption spectroscopy, line 238 and following) has to be verified, the indicated wavelengths were used to distinguish morphine and codeine, the general color change was from red to blue (520/690).

 Response: Thank you so much for pointing out. We have taken care.

However, the modifications suffer from the limited time the Authors had available to provide so substantial text changes.

The first version of the text was prepared practically without noticeable language errors. The current version must be corrected (in the parts that were modified) as there are numerous errors (grammar) and in some cases the units of measure were affected (for example lines 243-244).

The number of errors is rather high, and I marked some of them in the text instead of preparing the list.

Response: We have changed all the marked corrections and other typo errors and figure. We appreciate that reviewer has pointed out and checked these minute errors. We have made the changes in the revised version. Beside this, we have taken the help of native English speaker to check the grammar and language.

We have sincerely, read the manuscript and taken care of all the punctuation and grammar mistakes with more diligently and care.

Thank you so much for diligently review.
